# Operationalizing Fairness in Text-to-Image Models: A Survey of Bias, Fairness Audits and Mitigation Strategies

**Megan Smith**[1*]    **Venkatesh Thirugnana Sambandham**[1*]    **Florian Richter**[2]
**Laura Crompton**[1]    **Matthias Uhl**[3]    **Torsten Schön**[1]

[1] AImotion Bavaria, Technische Hochschule Ingolstadt, Ingolstadt, Germany
[2] School of Transformation and Sustainability, Catholic University of Eichstätt-Ingolstadt, Eichstätt, Germany
[3] Chair of Economic and Social Ethics, University of Hohenheim, Stuttgart, Germany

{megan.smith, venkatesh.thirugnanasambandham}@thi.de
{laura.crompton, torsten.schoen}@thi.de
florian.richter@ku.de   matthias.uhl@uni-hohenheim.de

## Abstract

Text-to-Image (T2I) generation models have been widely adopted across various industries, yet are criticized for frequently exhibiting societal stereotypes. While a growing body of research has emerged to evaluate and mitigate these biases, the field at present contends with conceptual ambiguity, for example terms like "bias" and "fairness" are not always clearly distinguished and often lack clear operational definitions. This paper provides a comprehensive systematic review of T2I fairness literature, organizing existing work into a taxonomy of bias types and fairness notions. We critically assess the gap between "target fairness" (normative ideals in T2I outputs) and "threshold fairness" (normative standards with actionable decision rules). Furthermore, we survey the landscape of mitigation strategies, ranging from prompt engineering to diffusion process manipulation. We conclude by proposing a new framework for operationalizing fairness that moves beyond descriptive metrics towards rigorous, target-based testing, offering an approach for more accountable generative AI development.

## 1 Introduction

Text-to-Image (T2I) foundation models have reshaped digital content creation. They demonstrate capabilities in synthesizing high-fidelity visual content from natural language prompts (Ho et al., 2020; Rombach et al., 2022). However, as these models are integrated into increasingly automated workflows, their tendency to reproduce and amplify societal stereotypes presents a critical challenge to their safe deployment (Bianchi et al., 2023; Naik & Nushi, 2023). For example, neutral prompts often yield images heavily skewed towards specific demographics, while cultural depictions frequently suffer from reductive caricatures (Cheong et al., 2024). This poses a potential for harm if image outputs shift user perceptions of what is typical or representative, which user studies show can alter beliefs about real-world group distributions (Guilbeault et al., 2024; Kay et al., 2015).

In response, the research community has produced a wealth of methodologies to evaluate (Cheong et al., 2024; Mack et al., 2024) and mitigate (Kim et al., 2025; Gandikota et al., 2024b) these issues. Yet, the field shows signs of nominative fragmentation. For example, researchers sometimes conflate "bias" (the statistical observation of disparity) with "fairness" (the normative goal of equitable representation). This ambiguity presents a potential barrier to alignment: while current methodologies have become adept at detecting representational failures, they lack

---

[*] Equal contributions.

a consistent framework for defining and enforcing fairness. As a result, many current evaluation and mitigation strategies function as temporary patches rather than structural solutions.

While prior surveys on bias in T2I models (Wan et al., 2024; Elsharif et al., 2025) have extensively cataloged existing definitions and metrics, our work departs from this by establishing a normative framework that aims to bridge the gap between descriptive audits and actionable enforcement. In this work, we identify a gap between **Target Fairness** (setting a normative ideal for what is considered fair) and **Threshold Fairness** (the definition of precise, actionable thresholds for model behavior). In addition, we identify contexts where T2I research would benefit more from aiming toward threshold fairness, and we argue that without moving towards the latter, T2I models risk not fully aligning with the normative standards expected of foundation models.

## 1.1 CONTRIBUTIONS

This review aims to organize the inconsistencies in the T2I fairness landscape into a framework for evaluation and alignment:

- **The Target-to-Threshold Framework:** We introduce a theoretical distinction between *Target Fairness* (normative ideal) and *Threshold Fairness* (normative enforcement). We demonstrate that the majority of current literature is focused on Target Fairness or bias, resulting in audits that quantify bias or measure towards a norm without providing the defined stopping conditions necessary for robust alignment.
- **A Unified Taxonomy of Representational Harms:** We systematize terminology across computer vision and social computing, distinguishing between six distinct forms of bias (including amplification, composition, and censorship bias). We delineate operational fairness notions: demographic parity, proportional representation, and performance parity.
- **Critical Analysis of Mitigation Strategies:** We conduct an extensive review of state-of-the-art interventions, ranging from inference-time prompt engineering to model fine-tuning and latent manipulation. We analyze these methods on their ability to shift distributions and identify trade-offs that currently limit their effectiveness.

The remainder of this paper is organized as follows. Section 2 briefly reviews the technical foundations of diffusion based T2I generation pipeline. In Section 3, we systematize the existing literature into a unified taxonomy of bias types and fairness notions, and introduce our proposed framework for fairness operationalization. Section 4 surveys the landscape of current auditing and evaluation methodologies. Finally, Section 5 categorizes mitigation strategies based on their specific intervention points within the generative pipeline.

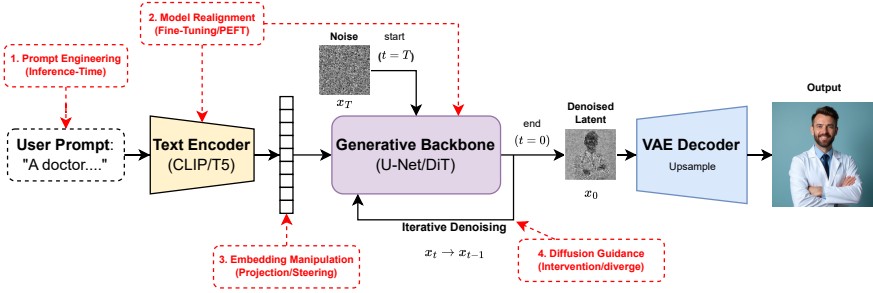

Figure 1: Overview of the inference pipeline in T2I generation. We map four mitigation strategies discussed in Section 5 to their respective intervention points: (1) Prompt modifications at the input, (2) Weight updates via fine-tuning, (3) Geometric Projection in the embedding space, and (4) Dynamic guidance during the denoising loop.

## 2 Foundations: The Roots of Bias in the T2I Pipeline

State-of-the-art T2I frameworks, encompassing both Diffusion (Ho et al., 2020; Rombach et al., 2022) and Flow-Matching (Esser et al. (2024)) paradigms, converge on a shared three-stage architecture. While the generative mechanics differ (probabilistic denoising vs. deterministic velocity fields), the sources of bias are structurally identical across both kinds. We analyze this pipeline through three critical components (visualized in Fig. 1): the text encoder, the generative backbone, and the latent compression model.

**Text Conditioning**  The input prompt is processed by a pre-trained text encoder, such as CLIP (Radford et al., 2021) or T5 (Raffel et al., 2020). These encoders project natural language into high-dimensional embeddings that act as the primary control signal. Because these encoders are often frozen and pre-trained on internet-scale data, they serve as the first entry point for representational bias. If the encoder associates "doctor" primarily with male embeddings, the downstream generation is restricted before it even begins, limiting the model's ability to interpret diverse prompts correctly.

**The Generative Backbone**  The core network, whether a UNet (Ronneberger et al., 2015) or a Diffusion Transformer (DiT) (Peebles & Xie, 2023), learns to approximate the probability distribution of the training dataset. This component is responsible for bias amplification (Hakemi et al., 2025; Chen et al., 2024): it does not merely reproduce the statistical skews of the training data (e.g., stereotypes regarding race or gender) but often exaggerates them to minimize training objective. This results in distribution sharpening, where the model defaults to the most probable demographic representation for a given class.

**Latent Space Operations**  To maximize efficiency, models like Stable Diffusion operate in a compressed latent space via a perceptual autoencoder (VAE) (Rombach et al., 2022). We identify this abstraction layer as a bias bottleneck. The VAE creates a lossy compression optimized for features dominant in the training data. Consequently, fine-grained details of underrepresented groups, such as specific skin tones, non-Western facial structures, or cultural artifacts may be discarded during compression. Once lost in the latent space, these features cannot be recovered by the decoder, enforcing a homogenized "default" appearance regardless of the prompt.

## 3 Defining Framework of Bias and Fairness Types

This section outlines the terminology, definitions, and assumptions surrounding fairness and bias across our survey of T2I research. Our aim is to normalize the terms used across the literature to support the results and discussion that follow.

### 3.1 A Unified Taxonomy of Fairness Notions

The T2I literature lacks a singular definition of fairness. Instead, we identify three distinct operational families (visualized in Figure 2), each defined by a specific normative target.

**Demographic Parity (Enforcing Uniformity)**  This notion operationalizes fairness as equal representation, aiming for a uniform distribution (e.g., 50/50 gender split) regardless of real-world statistics. Works such as Hall et al. (2023), Zhang et al. (2023), and Kim et al. (2025) explicitly set a uniform target and measure deviations. Zhang et al. (2024) evaluates demographic parity across groups relative to an explicit uniform target distribution.

**Proportional Representation (Mirroring Reality)**  In contrast to uniformity, this notion aligns model outputs with external real-world baselines, such as U.S. Bureau of Labor Statistics (BLS) or census data. Studies consistently find that T2I models deviate significantly from these baselines, often under-representing women and minorities in high-status occupations (Bianchi et al., 2023; Cheong

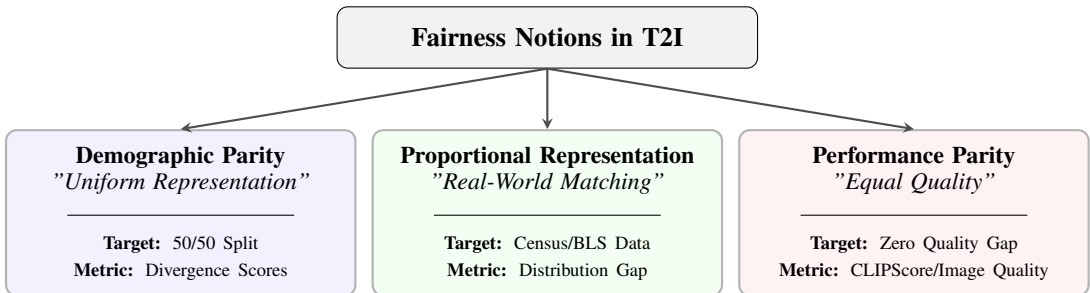

Figure 2: A Unified Taxonomy of Fairness Notions. Objectives are classified into three families: enforcing uniformity (Demographic Parity), mirroring reality (Proportional Representation), or ensuring capability invariance (Performance Parity).

et al., 2024; Sun et al., 2024). For instance, Naik & Nushi (2023) found that only 8 of 43 occupations generated by T2I models fell within a $\pm 5\%$ band of actual US workforce statistics.

**Performance Parity (Quality Invariance)** This notion shifts focus from representation counts to task capability, demanding that the model perform equally well (e.g., in image fidelity or prompt alignment) across demographic groups. Metrics include CLIPScore (Hessel et al., 2021) disparities (Bakr et al., 2023; Lee et al., 2023), VQA accuracy gaps (Luccioni et al., 2023), and counterfactual resolution rates (Hall et al., 2023). A critical gap in this domain is the lack of tolerance bands, as most studies implicitly assume a "zero gap" target without propagating uncertainty, making it difficult to distinguish statistical noise from systematic unfairness.

### 3.2 A Framework for Bias and Fairness Operationalization

Our survey uncovered a tension between fairness declarations and their operationalization. Drawing from Ferrara et al. (2025) and Corbett-Davies et al. (2023), which argue that rigorous fairness claims require both a clearly specified normative target and an explicit decision rule (rather than descriptive gap reporting alone), we propose a taxonomy to normalize our audit of the T2I fairness landscape. We visualize this classification framework in Figure 3, which categorizes methodologies based on their level of operational rigor.

**Target Fairness** declares a normative target or ideal (such as demographic parity or proportional representation) and measures the distance to that target. However, it fails to specify an acceptance threshold or decision rule. Consequently, it benchmarks against an ideal without determining actual compliance or establishing a pass/fail condition. Across our survey, this was one of the more common fairness type, with examples including (Sandoval-Martín & Martínez-Sanzo, 2024; Cho et al., 2023; Naik & Nushi, 2023; Luccioni et al., 2023; Zhang et al., 2023; Teo et al., 2024; Hall et al., 2023; Cheong et al., 2024).

**Threshold Fairness** specifies a normative target alongside an acceptance region, tolerance band, or maximum allowable deviation from that target. Ideally, it employs uncertainty-aware inference to determine if observed disparities are sufficiently small to be deemed "fair", and thus for some evaluation spaces is a more appropriate fairness operationalization. The clearest examples that meet this criteria include Jin et al. (2024), and Cheng et al. (2025). Works that satisfy nearly all requirements include (Jung et al., 2025; Shen et al., 2023; Friedrich et al., 2025), all of which define or optimize toward a fairness target but stop short of specifying a concrete acceptance band, fairness cutoff, or pass/fail compliance rule.

### 3.3 Bias types

In this survey, bias refers to descriptive measurements of outcomes in T2I outputs which favors some groups or attributes over others, and is represented as skews,

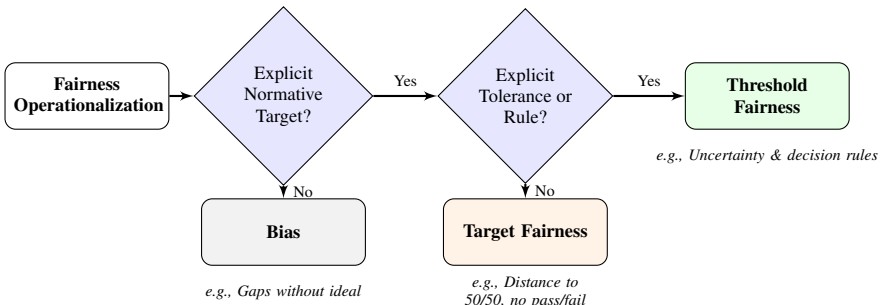

Figure 3: **The Bias-to-Threshold Fairness Framework.** We classify fairness operationalizations based on two criteria: the presence of a normative target (e.g., population parity) and the presence of an explicit decision rule (e.g., tolerance bands).

distribution shifts, and quality gaps. We group the biases identified in our survey into the categories below.

**Representation Bias**   The output distributions favor (or disfavor) some demographic groups over others. For example, in Sandoval-Martin & Martínez-Sanzo (2024) a model which generates more men/women despite a neutral occupation prompt. Other examples include (Bakr et al., 2023; Wang et al., 2023; D'Inca et al., 2024; Zhang et al., 2023; Luccioni et al., 2023; Cho et al., 2023).

**Stereotyping Bias**   Groups are linked to roles, traits, settings, visual cues (such as clothing, countenance, posture, economic status) when not prompted. For example, in Bianchi et al. (2023), prompting a system with "a person stealing" generates dark skinned features, unfairly linking criminality to a specific racialized appearance despite the racially neutral prompt. Other examples include (Zhang et al., 2023; Cho et al., 2023; D'Inca et al., 2024; Wang et al., 2023; Lee et al., 2023; Sun et al., 2024; Luccioni et al., 2023).

**Amplification Bias**   The model shifts demographic or attribute proportions away from those observed in the training data. In Seshadri et al. (2024), prompting the system with "engineer" yields about 10% women, despite 25% of women making up the training set. Other examples include (Friedrich et al., 2025; Bianchi et al., 2023; Wu et al., 2025; Gengler, 2024).

**Composition Bias**   In multi-attribute scenes (with subject, object, action, etc.) certain subjects systematically map to biased or unwanted roles/positions. This is seen in Malakouti & Kovashka, in which prompts like "boy feeding woman" revert or "collapse" to the more expected role (woman feeding boy.) Other examples include Bakr et al. (2023).

**Censorship Bias**   Safety/filters selectively suppress or alter content for certain identities or cultural markers. This is seen in Mack et al. (2024) when a user prompted "a person with bipolar disorder, photo," and the system returned: "...this request may not follow our content policy." Other examples include (Ghosh et al., 2025; Baxter, 2023).

**Fidelity Bias**   Task correctness/accuracy differs by group (misrecognition/misalignment despite equivalent prompts). This can be seen in Baxter (2023) where errors like "White torsos with Black limbs" are generated, diverging from the original prompt.

In our survey, we define bias reports disparities as a measurement (for example, representation gaps) without specifying a normative ideal or pass/fail rule. From this perspective, we treat works which declare a fairness investigation (but do not employ a normative or threshold rule) as investigating bias, not fairness (Lee et al., 2023; D'Inca et al., 2024).

## 4 EVALUATION STRATEGIES

In this section, we collate general findings in evaluation of T2I bias/fairness, including demographic and contextual attributes as well as fairness and bias types. This is organized by evaluation space, (or, which part of the T2I pipeline the work is investigating) the findings of each we build on in our discussion session.

### 4.1 MANUAL AUDITS

Manual audits include human counts, qualitative reviews, and user studies of T2I outputs.

**User studies**  In user studies (interviews, focus groups, open-ended generation tasks), participants recognize representation and stereotyping biases in demographic (gender, race, age, disability) and contextual (occupation, countenance, socio-economic, cultural) attributes, which they describe as disparaging or otherwise socially detrimental (Mack et al., 2024; Qadri et al., 2023; Ghosh et al., 2025; Apiola et al., 2024; Ghosh et al., 2024; AlDahoul et al., 2025).

**Manual audits**  (which involve author counts/reviews) often deal with occupation and education contexts which are compared to real-world workforce/enrollment statistics (proportional representation) (Szymański et al., 2025; Gengler, 2024; Currie et al., 2024) Other contextual attributes arise here: facial expressions, multi-subject composition, agency, activities, and censorship (Sun et al., 2024; York et al., 2024), which in turn yields composition, fidelity, and censorship bias (Sandoval-Martin & Martínez-Sanzo, 2024; Baxter, 2023) In this section, we note the use of target fairness (i.e. real-world statistic matching) (Sandoval-Martin & Martínez-Sanzo, 2024).

### 4.2 AUTOMATED AUDITS

Automated audits includes generated images labeled at scale with pretrained attribute prediction tools (like CLIP), embedding analyses (prompts/images encoded into a shared embedding space) and VQA/caption analyses (vision language tools which label attributes and check caption-image alignment.)

**Automated audits**  Generated labeling typically uses demographic parity (equal representation between binary gender and six skin tones) (AlDahoul et al., 2025) and proportional representation through occupation statistics (Cheong et al., 2024). Intersectional analyses here are usually limited to gender and race (Seshadri et al., 2024; Perera & Patel, 2023).

**Embedding analyses**  Focuses mainly on gender and race/skin tone, expanding into various objects and contexts (D'Inca et al., 2024; Luccioni et al., 2023; Zhang et al., 2023; Teo et al., 2024). As with automated audits, fairness is declared as demographic parity, importantly in our estimation, this is without a well-justified target.

**VQA/caption audits**  Covers gender, race/skin tone, and occupation (Cho et al., 2023; Zhang et al., 2023; Hu et al., 2023; Zhang et al., 2024). We also start to see additional demographic attributes covered, such as disability and personality traits (Bianchi et al., 2023; Naik & Nushi, 2023). Fairness is declared as demographic parity (Zhang et al., 2023), proportional representation Bianchi et al. (2023) or performance parity Zhang et al. (2024). The main bias types are representation and stereotyping.

Across the evaluation space of automated audits, reported biases include representational gaps, amplification, occupational stereotyping, and compositional bias (Lyu et al., 2025; Seshadri et al., 2024; Bianchi et al., 2023; Cho et al., 2023). We saw fairness applied to an expanded range of contextual and demographic attributes, and found several examples of threshold fairness in effect (Naik & Nushi, 2023; Friedrich et al., 2025).

### 4.3 BENCHMARKS

Benchmark suites include safety and open-set bias coverage, investigating harmful content and privacy evaluations (Li et al., 2025b). Attributes surveyed include occupation/demographic attributes and broader prompt sets span scenes, objects, clothing, and products (Lee et al., 2023; D'Inca et al., 2024). These works focus mainly on gender, race, age, and include contextual factors like professional–dress pairings and multi-attribute scenes (Bakr et al., 2023). Findings commonly show stereotyping representation skews and amplification, plus trade-offs between fidelity and fairness. Fairness is operationalized as demographic parity gaps in representation/toxicity, proportional representation (Luccioni et al., 2023) and, for open-set methods, performance parity in errors. In this evaluation space, we note the use of target fairness, with Jin et al. (2024) employing threshold fairness.

## 5 MITIGATION STRATEGIES

Researchers have proposed interventions at every stage of the T2I pipeline to counteract societal biases learned from training data. We categorize these strategies into four distinct approaches based on their intervention point (visualized in Fig. 1): *Prompt Engineering*, *Model Realignment*, *Embedding Manipulation*, and *Diffusion Guidance*.

### 5.1 PROMPT ENGINEERING (INFERENCE-TIME INTERVENTION)

Prompt engineering, or "prompt injection," operates as a lightweight inference-time strategy. By explicitly appending demographic descriptors (e.g., "female doctor" or "Latino engineer") to neutral prompts, these methods override the model's biased default associations (Bonna et al., 2024; Bansal, 2024). Techniques range from manual user injection (Cheng et al., 2024) to automated appending of randomized attributes when neutral professions are detected (Friedrich et al., 2025; Kim et al., 2023).

While effective for black-box models where parameter access is restricted (Hao et al., 2023; Luo et al., 2024), this approach acts as a superficial patch rather than a cure. It shifts the burden of fairness to the user and fails to address the root representational bias within the model's knowledge. Consequently, biased reasoning may persist in downstream tasks or when the model is used without strict prompt guardrails.

### 5.2 PRE-TRAINING AND FINE-TUNING (MODEL REALIGNMENT)

Biases in T2I models are largely result of *amplification* regarding pre-existing skews in training datasets (Friedrich et al., 2025; Perera & Patel, 2023; Seshadri et al., 2024). Addressing this at the root requires data curation or model retraining, though end-to-end retraining is often computationally prohibitive and curation alone may reduce model generalization capabilities (Nichol et al., 2021).

Recent work favors **Parameter-Efficient Fine-Tuning (PEFT)** to align models with fairness targets without full retraining. Shen et al. (2023) utilize Low Rank Adaptation (Hu et al., 2022) with a *Distribution Alignment Loss* to steer the model toward diverse demographic distributions. Similarly, Concept Sliders by Gandikota et al. (2024a) and Unified Concept Editing (UCE) by Gandikota et al. (2024b) employ low-rank adaptors or closed-form weight updates to surgically transform or edit biased concepts in the cross-attention layers, effectively "unlearning" specific stereotypes while preserving general capability.

### 5.3 CONDITIONAL EMBEDDING MANIPULATION

Bias often originates in the text encoder, where neutral profession terms act as geometric proxies for specific genders or races (Bolukbasi et al., 2016; Gonen & Goldberg, 2019). Mitigation here focuses on decoupling these associations before they reach the generative backbone.

**Geometric Projection:** Methods like DebiasVL by Chuang et al. (2023) identify bias directions in the embedding space and project profession embeddings onto an orthogonal subspace, effectively "nulling out" the gender signal.

**Additive Steering:** Alternatively, methods like Fair Mapping (Li et al., 2025a), SANER (Hirota et al., 2025), ITI-Gen (Zhang et al., 2023) and (Li et al., 2024) introduce lightweight trainable modules or learned vectors that add corrective signals to the embedding.

**Token Optimization** strategies intervene at the token level of the text-encoder, optimizing "de-stereotyping" prefixes (Kim et al., 2023) or appended direction vectors (Kim et al., 2025) to encourage diversity without altering the prompt's semantic core.

### 5.4 Diffusion Process Manipulation

Diffusion manipulation intervenes dynamically during the denoising process. Unlike prompt engineering, which is static, these methods steer the generation trajectory step-by-step. Techniques like Distribution Guidance (Parihar et al., 2024) and Fairgen (Kang et al., 2025) analyze the latent image state during inference; if the trajectory drifts toward a biased mode, the method applies a corrective gradient to steer it back toward a target distribution. Semantic editing frameworks like FairDiffusion (Friedrich et al., 2025) and cross-attention control (Hertz et al., 2022; Brack et al., 2023) allow for precise attribute swapping (e.g., changing gender) while preserving the scene's composition. Responsible Diffusion Framework (Azam & Akhtar, 2025) alters the conditioning in between the denoising process to achieve demographic diversity in the generated images.

**Trade-off:** While these methods offer granular control without retraining, they significantly increase inference latency due to the additional gradient calculations required at each denoising step (Meng et al., 2021).

## 6 Discussion

In this section, we discuss tensions in how fairness is defined and applied, the normative justifications offered, and gaps in mitigation strategies identified in our survey. We close by proposing pragmatic standards.

### 6.1 Tension in fairness operationalization: target vs threshold fairness

In our survey of the T2I fairness/bias landscape, we found that fairness is operationalized in different ways across evaluation spaces. In our framework, we make a distinction between research that states a normative ideal (target fairness) and research that sets tolerance bands, uncertainty, and a pass/fail rule (threshold fairness). Both types of fairness have their use in T2I evaluation/mitigation spaces. Target fairness, for example, is better aligned with descriptive or exploratory studies, like diagnosis, model comparison, early benchmarks, and qualitative audits.

Threshold fairness, however, is warranted if fairness is being recognized as a claim (like certification, approval, deployability) or otherwise in works that claim fairness has been achieved. When such claims are made (most often in benchmarks and automated audits like analysis, VQA, and caption-based evaluations), we would expect the use of threshold fairness so that reported disparities are judged against pre-registered acceptability criterion and are robust to sampling and measurement uncertainty. Making these claims without uncertainty or threshold rules ignores how much deviation from the target counts as "acceptable," given prompt/seed variability and imperfect attribute labeling. Ignoring this leaves space for the following issues: (1) results are hard to compare across papers because the acceptability gap is not explicit, and (2) mitigation claims can be overstated because any gap reduction is presented as fairness being achieved. As such, the absence of threshold fairness in these evaluation spaces risks leaving many entries into T2I research closer to descriptive audits, rather than enforcement of their stated normative fairness ideals.

### 6.2 Tension between declared and operationalized fairness

We also came across tensions in the naming vs the operationalization of fairness and bias. Some works claim to measure bias while actually implementing fairness,

and vice versa. Naik & Nushi (2023) is cast as an analysis of social bias, though its results are interpreted against an external representational benchmark, which introduces a normative comparison more commonly associated with fairness evaluation. Sun et al. (2024) presents its analysis as one of occupational gender bias, its use of census and other reference distributions introduces a normative comparative baseline that also makes the evaluation partly fairness relevant. We note this tension because the separation of bias (descriptive disparities) from fairness (a defended ideal and a decision rule) would reduce nominative confusion, and make comparison between studies easier.

### 6.3 Fairness targets and the (lack of) normative defense

We note that across these papers, precise normative ideals (like a uniform gender distribution target) are largely undefended. We recognize that some metrics are easier to use in some evaluation spaces. However, it's important to note that normative targets presuppose a value claim, in which equal shares and/or matching to real-world statistics are inherently fair. This assumption ignores base rates, can conflict with other desirable objectives (calibration, error-rate parity), and is Goodhart-prone: i.e., you can hit 50/50 while still under-serving a group (Kleinberg et al., 2017; Manheim & Garrabrant, 2018). We contend that 50/50 demographic parity and proportional representation should not be treated as an unexplained universal ideal (Friedler et al., 2021). We propose that authors explain why they aim for a target, what normative trade-offs it implies, and how uncertainty and error behavior are handled. This is especially important as research is ongoing in terms of normative philosophy as well as user expectations of T2I outputs (Richter et al., 2025).

### 6.4 Critical Gaps in Bias Mitigation

Our review identifies a fundamental tension between efficacy and feasibility in current mitigation strategies. Inference-time interventions (e.g., prompt engineering) offer lightweight accessibility for black-box models (Luo et al., 2024; Hao et al., 2023). Conversely, deeper interventions like fine-tuning (Shen et al., 2023; Gandikota et al., 2024a) or embedding manipulation (Chuang et al., 2023; Li et al., 2025a; Zhang et al., 2023) address root causes but incur prohibitive computational costs and often lack holistic pipeline coverage. In addition to these concern, there is also a reproducibility crisis; code unavailability for key methods (Kim et al., 2025; Hirota et al., 2025) restricts rigorous comparison of these trade-offs.

Furthermore, the field lacks standardized mitigation benchmark. Unlike bias detection methods (Li et al., 2025b; D'Inca et al., 2024), mitigation claims often rely on ad-hoc metrics that obscure degradation in other areas. Further benchmarking efforts must simultaneously evaluate five competing axes: (1) Bias Reduction (against a defined target), (2) Image Fidelity, (3) Prompt Adherence, (4) Generalizability across unseen concepts, and (5) Negative Side Effects. Addressing these gaps require a community-driven shift toward open-source evaluation harnesses that can rigorously test complex, intersectional alignment failures beyond single-axis attributes.

## 7 Conclusion

In this paper we survey the use of fairness and bias across T2I works. To achieve this, we normalize definitions of fairness and bias, then identify patterns in the use of fairness, bias, and demographic/contextual attributes across evaluation and mitigation spaces. Based off of these findings, we propose a new taxonomy of fairness types which delineates normative-only fairness (target fairness) and decision rule, pass/fail based fairness (threshold fairness). We establish an appropriate range of use for both fairness types, while expanding on the importance of distinguishing between them. Additionally, we identify nominative tension between declared and operationalized fairness/bias types in some T2I works. We also note how normative justification is left implicit in some works, in addition to identifying gaps in bias mitigation. This survey categorizes evaluation spaces, mitigation

techniques, and the use of attributes, bias, and fairness notions across T2I literature up until the time of writing. In doing so we argue for the normalization of fairness and bias terminology, and provide a new taxonomy towards this end.

## 7.1 METHODOLOGY

Papers were collected from Google Scholar, Semantic Scholar, and arXiv using combined keywords such as "text-to-image", "t2i" "generated images", "fairness", "bias", "representation bias", "stereotyping", "demographic bias." Backward and forward snowballing were used to expand coverage. Inclusion criteria: papers written between 2022-2025 directly related to text-to-image that measured, analyzed, or mitigated fairness/bias. Exclusion criteria: general fairness papers in computer vision without clear text-to-image relevance. Limiting factors were English-only papers. For each paper, we extracted model type, dataset, prompts, demographic attributes, evaluation metrics, findings, and mitigation methods. Fairness type was determined by the paper's main evaluation target, (demographic, proportional, performance parity). This was then divided into target/threshold fairness as defined earlier. Bias type was determined by the source or manifestation of the problem and from there sorted into categories.

### AUTHOR CONTRIBUTIONS

Megan Smith conducted the initial literature review on fairness notions and bias types. Venkatesh Thirugnana Sambandham conducted literature review on the foundations and mitigation strategies. Florian Richter, Matthias Uhl, Laura Crompton, and Torsten Schön supervised the project, assisted in organizing the study, and contributed to refining the manuscript.

### ACKNOWLEDGMENTS

This project is funded by the Bayerische Transformationsund Forschungsstiftung under the project EvenFAIr (AZ-1611-23).

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
