# OpenReview forum: "Operationalizing Fairness in Text-to-Image Models: A Survey of Bias, Fairness Audits and Mitigation Strategies"
_ICLR.cc/2026/Workshop/AFAA — AFAA 2026 Oral_

### Official Review · Reviewer_9BVf · 2026-02-21
**The paper introduces a new framework to address conceptual fragmentation in T2I bias research**

**Rating:** 4
**Confidence:** 4

**Summary:**

This paper makes a timely and well-structured contribution by introducing a Target-to-Threshold fairness framework that addresses genuine conceptual fragmentation in T2I bias research.

**Strengths:**

The paper makes a genuinely useful conceptual contribution by distinguishing Target Fairness from Threshold Fairness, filling a real gap in how the field conflates descriptive auditing with normative enforcement. The taxonomy of six bias types is well organized, and the coverage of mitigation strategies across pipeline intervention points is systematic and technically grounded.

**Weaknesses:**

The survey also skews heavily toward Western models and datasets, with limited engagement with non-Western cultural contexts, despite fairness being a deeply culturally contingent concept.

---

### Official Review · Reviewer_ZTPL · 2026-02-21

[review text omitted: it was posted to a different submission]

---

### Official Review · Reviewer_Z9gr · 2026-02-21
**Nice paper and overview**

**Rating:** 4
**Confidence:** 4

**Summary:**

This paper provides a comprehensive systematic review of T2I fairness literature,
organizing existing work into a taxonomy of bias types and fairness notions.

**Strengths:**

In terms of what a conference submission can provide, this paper does a great job giving a summary of the current literature in machine learning approaches for text to image models and the issues around metrics and strategies to ensure fairness.  In particular, the inherent tension between  between ”target fairness” (normative ideals in T2I outputs)
and ”threshold fairness” (normative standards with actionable decision rules) is an important piece to highlight.

**Weaknesses:**

None

---

### Meta-Review · Area_Chair_hbAT · 2026-02-23

**Recommendation:** Main Papers Track
**Confidence:** 4

**Metareview:**

The paper presents a timely and well-written taxonomy of types of bias and definitions of fairness in T2I generation models. All authors agree that this would be a valuable contribution to the workshop. I encourage the authors to incorporate the reviewers' suggestions, in particular on discussing the value of this current taxonomy as compared to similar ones.

---

### Decision · Program_Chairs · 2026-03-02

Accept (Oral)